# Can Word Sense Distribution Detect Semantic Changes of Words?

**Xiaohang Tang**[†]  **Yi Zhou**[♣]   **Taichi Aida**[◇]  **Procheta Sen**[†]  **Danushka Bollegala**[†,‡]

University of Liverpool, United Kingdom.[†]
Cardiff University, United Kingdom.[♣]
Tokyo Metropolitan University, Japan.[◇]
Amazon[‡]

xiaohangtang01@gmail.com zhouy131@cardiff.ac.uk
aida-taichi@ed.tmu.ac.jp {procheta.sen,danushka}@liverpool.ac.uk

## Abstract

Semantic Change Detection (SCD) of words is an important task for various NLP applications that must make time-sensitive predictions. Some words are used over time in novel ways to express new meanings, and these new meanings establish themselves as novel senses of existing words. On the other hand, Word Sense Disambiguation (WSD) methods associate ambiguous words with sense ids, depending on the context in which they occur. Given this relationship between WSD and SCD, we explore the possibility of predicting whether a target word has its meaning changed between two corpora collected at different time steps, by comparing the distributions of senses of that word in each corpora. For this purpose, we use pretrained static sense embeddings to automatically annotate each occurrence of the target word in a corpus with a sense id. Next, we compute the distribution of sense ids of a target word in a given corpus. Finally, we use different divergence or distance measures to quantify the semantic change of the target word across the two given corpora. Our experimental results on SemEval 2020 Task 1 dataset show that word sense distributions can be accurately used to predict semantic changes of words in English, German, Swedish and Latin.

## 1 Introduction

SCD of words over time has provided important insights for diverse fields such as linguistics, lexicography, sociology, and information retrieval (IR) (Traugott and Dasher, 2001; Cook and Stevenson, 2010; Michel et al., 2011; Kutuzov et al., 2018). For example, in IR one must know the seasonal association of keywords used in user queries to provide relevant results pertaining to a particular time period. Moreover, it has been shown that the performance of publicly available pretrained LLMs declines over time when applied to emerging data (Su et al., 2022; Loureiro et al., 2022; Lazaridou et al., 2021) because they are trained

using a static snapshot. Moreover, Su et al. (2022) showed that the temporal generalisation of LLMs is closely related to their ability to detect semantic variations of words.

A word is often associated with multiple *senses* as listed in dictionaries, corresponding to its different meanings. Polysemy (i.e. coexistence of several possible meanings for one word) has been shown to statistically correlate with the rate of semantic change in prior work (Bréal, 1897; Ullmann, 1959; Magué, 2005). For example, consider the word *cell*, which has the following three noun senses according to the WordNet[1]: (a) **cell%1:03:00** – *the basic structural and functional unit of all organisms*, (b) **cell%1:06:04** – *a handheld mobile radiotelephone for use in an area divided into small sections*, and (c) **cell%1:06:01** – *a room where a prisoner is kept*. Here, the WordNet sense ids for each word sense are shown in boldface font. Mobile phones were first produced in the late 1970s and came into wider circulation after 1990. Therefore, the sense (b) is considered as a more recent association compared to (a) and (c). Given two sets of documents, one sampled before 1970 and one after, we would expect to encounter (b) more frequently in the latter set. Likewise, articles on biology are likely to contain (a). As seen from this example **the sense distributions of a word in two corpora provide useful information about its possible meaning changes over time**.

Given this relationship between the two tasks SCD and WSD, a natural question arises – *is the word sense distribution indicative of semantic changes of words?* To answer this question, we design and evaluate an unsupervised SCD method that uses only the word sense distributions to predict whether the meaning associated with a target word $w$ has changed from one text corpora $\mathcal{C}_1$ to another $\mathcal{C}_2$. For the ease of future references, we name this method Sense-based Semantic Change

---

[1]https://wordnet.princeton.edu/

Score (SSCS). Given a target word $w$, we first disambiguate each occurrence of $w$ in each corpus. For this purpose, we measure the similarity between the contextualised word embedding of $w$, obtained from a pre-trained Masked Language Model (MLM), from the contexts containing $w$ with each of the pre-trained static sense embeddings corresponding to the different senses of $w$. Next, we compute the sense distribution of $w$ in each corpus separately. Finally, we use multiple distance/divergence measures to compare the two sense distributions of $w$ to determine whether its meaning has changed from $C_1$ to $C_2$.

To evaluate the possibility of using word senses for SCD, we compare the performance of SSCS against previously proposed SCD methods using the SemEval-2020 Task 1 (unsupervised lexical SCD) (Schlechtweg et al., 2020) benchmark dataset. This task has two subtasks: (1) a *binary classification* task, where for a set of target words, we must decide which words lost or gained sense(s) from $C_1$ to $C_2$, and (2) a *ranking* task, where we must rank target words according to their degree of lexical semantic change from $C_1$ to $C_2$. We apply SSCS on two pre-trained static sense embeddings, and six distance/divergence measures. Despite the computationally lightweight and unsupervised nature of SSCS, our experimental results show that it surprisingly outperforms most previously proposed SCD methods for English, demonstrating the effectiveness of word sense distributions for SCD. Moreover, evaluations on German, Latin and Swedish show that this effectiveness holds in other languages as well, although not to the same levels as in English. We hope our findings will motivate future methods for SCD to explicitly incorporate word sense related information. Source code implementation for reproducing our experimental results is publicly available.[2]

## 2 Related Work

**Semantic Change Detection:** SCD is modelled in the literature as the unsupervised task of detecting words whose meanings change between two given time-specific corpora (Kutuzov et al., 2018; Tahmasebi et al., 2021). In recent years, several shared tasks have been held (Schlechtweg et al., 2020; Basile et al., 2020; Kutuzov and Pivovarova, 2021), where participants are required to predict the

degree or presence of semantic changes for a given target word between two given corpora, sampled from different time periods. Various methods have been proposed to map vector spaces from different time periods, such as initialisation (Kim et al., 2014), alignment (Kulkarni et al., 2015; Hamilton et al., 2016), and joint learning (Yao et al., 2018; Dubossarsky et al., 2019; Aida et al., 2021).

Existing SCD methods can be broadly categorised into two groups: (a) methods that compare word/context clusters (Hu et al., 2019; Giulianelli et al., 2020; Montariol et al., 2021), and (b) methods that compare embeddings of the target words computed from different corpora sampled at different time periods (Martinc et al., 2020; Beck, 2020; Kutuzov and Giulianelli, 2020; Rosin et al., 2022). Rosin and Radinsky (2022) recently proposed a temporal attention mechanism, which achieves SoTA performance for SCD. However, their method requires additional training of the entire MLM with temporal attention, which is computationally expensive for large MLMs and corpora.

The change of the *grammatical profile* (Kutuzov et al., 2021; Giulianelli et al., 2022) of a word, created using its universal dependencies obtained from UDPipe (Straka and Straková, 2017), has shown to correlate with the semantic change of that word. However, the accuracy of the grammatical profile depends on the accuracy of the parser, which can be low for resource poor languages and noisy texts. Sabina Uban et al. (2022) used polysemy as a feature for detecting lexical semantic change discovery. The distribution of the contextualised embeddings of a word over its occurrences (aka. *sibling* embeddings) in a corpus has shown to be an accurate representation of the meaning of that word in a corpus, which can be used to compute various semantic change detection scores (Kutuzov et al., 2022; Aida and Bollegala, 2023). XL-LEXEME (Cassotti et al., 2023) is a supervised SCD method where a bi-encoder model is trained using WiC (Pilehvar and Camacho-Collados, 2019) dataset to discriminate whether a target word appears in different senses in a pair of sentences. XL-LEXEME reports SoTA SCD results for English, German, Swedish and Russian.

**Sense Embeddings:** Sense embedding learning methods represent different senses of an ambiguous word with different vectors. The concept of multi-prototype embeddings to represent word senses was introduced by Reisinger and Mooney (2010).

---

[2]https://github.com/LivNLP/
Sense-based-Semantic-Change-Prediction

This idea was further extended by Huang et al. (2012), who combined both local and global contexts in their approach. Clustering is used in both works to categorise contexts of a word that belong to the same meaning. Although the number of senses a word can take depends on that word, both approaches assign a predefined fixed number of senses to all words. To address this limitation, Neelakantan et al. (2014) introduced a non-parametric model, which is able to dynamically estimate the number of senses for each word.

Although clustering-based approaches can allocate multi-prototype embeddings to a word, they still suffer from the fact that the embeddings generated this way are not linked to any sense inventories (Camacho-Collados and Pilehvar, 2018). On the other hand, knowledge-based methods obtain sense embeddings by extracting sense-specific information from external sense inventories, such as the WordNet (Fellbaum and Miller, 1998) or the BabelNet[3] (Navigli and Ponzetto, 2012). Chen et al. (2014) extended word2vec (Mikolov et al., 2013) to learn sense embeddings using WordNet synsets. Rothe and Schütze (2015) made use of the semantic relationships in WordNet to embed words into a shared vector space. Iacobacci et al. (2015) used the definitions of word senses in BabelNet and conducted WSD to extract contextual information that is unique to each sense.

Recently, contextualised embeddings produced by MLMs have been used to create sense embeddings. To achieve this, Loureiro and Jorge (2019) created LMMS sense embeddings by averaging over the contextualised embeddings of the sense annotated tokens from SemCor (Miller et al., 1993). Scarlini et al. (2020a) proposed SenseEmBERT (Sense Embedded BERT), which makes use of the lexical-semantic information in BabelNet to create sense embeddings without relying on sense-annotated data. ARES (context-AwaRe EmbeddinS) (Scarlini et al., 2020b) is a knowledge-based method for generating BERT-based embeddings of senses by means of the lexical-semantic information available in BabelNet and Wikipedia. ARES and LMMS embeddings are the current SoTA sense embeddings.

## 3 Sense-based Semantic Change Score

SSCS consists of two steps. First, in §3.1, we compute the distribution of word senses associated with

---

[3] https://babelnet.org/

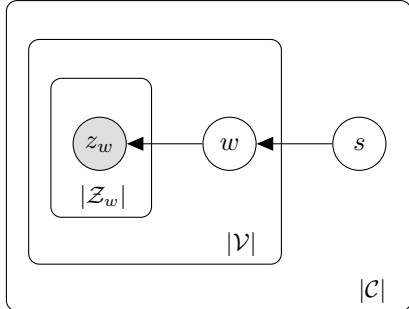

Figure 1: Plate diagram showing the dependencies among the sentences $s$ in the corpus $\mathcal{C}$, the target word $w$, and its sense $z$ in $s$.

a target word in a corpus. Second, in §3.2, we use different distance (or divergence) measures to compare the sense distributions computed for the same target word from different corpora to determine whether its meaning has changed between the two corpora.

### 3.1 Computing Sense Distributions

We represent the meaning expressed by a target word $w$ in a corpus $\mathcal{C}$ by the distribution of $w$'s word senses, $p(z_w|w, \mathcal{C})$. As explained in §1, our working hypothesis is that if the meaning of $w$ has changed from $\mathcal{C}_1$ to $\mathcal{C}_2$, then the corresponding sense distributions of $w$, $p(z_w|w, \mathcal{C}_1)$ will be different from $p(z_w|w, \mathcal{C}_2)$. Therefore, we first estimate $p(z_w|w, \mathcal{C})$ according to the probabilistic model illustrated in the plate diagram in Figure 1. We consider the corpus, $\mathcal{C}$ to be a collection of $|\mathcal{C}|$ sentences from which a sentence $s$ is randomly sampled according to $p(s|\mathcal{C})$. Next, for each word $w$ in vocabulary $\mathcal{V}$ that appears in $s$, we randomly sample a sense $z_w$ from its set of sense ids $\mathcal{Z}_w$. As shown in Figure 1, we assume the sense that a word takes in a sentence to be independent of the other sentences in the corpus, which enables us to factorise $p(z|w, \mathcal{C})$ as in (1).

$$p(z_w|w, \mathcal{C}) = \sum_{s \in \mathcal{C}(w)} p(z_w|w, s)p(s|\mathcal{C}) \quad (1)$$

Here, $\mathcal{C}(w)$ is the subset of sentences in $\mathcal{C}$ where $w$ occurs. We assume $p(s|\mathcal{C})$ to be uniform and set it to be $1/|\mathcal{C}|$, where $|\mathcal{C}|$ is the number of sentences in $\mathcal{C}$.

Following the prior works that use static sense embeddings for conducting WSD, the similarity between the pre-trained static sense embedding $\boldsymbol{z}_w$ of the sense $z_w$ of $w$, and the contextualised word embedding $\boldsymbol{f}(w, s)$ of $w$ in $s$ (obtained from

a pre-trained MLM) can be used as the confidence score for predicting whether $w$ takes the sense $z_w$ in $s$. Specifically, the LMMS and ARES sense embeddings we use in our experiments are computed using BERT (Devlin et al., 2019) as the back-end, producing aligned vector spaces where we can compute the confidence scores using the inner-product as given by (2).

$$p(z_w|w, s) = \frac{\langle \boldsymbol{z}_w, \boldsymbol{f}(w, s) \rangle}{\sum_{z'_w \in \mathcal{Z}_w} \langle \boldsymbol{z'}_w, \boldsymbol{f}(w, s) \rangle} \quad (2)$$

In WSD, an ambiguous word is typically assumed to take only a single sense in a given context. Therefore, WSD methods assign the most probable sense $z_w^*$ to $w$ in $s$, where $z_w^* = \arg\max_{z_w \in \mathcal{Z}_w} p(z_w|w, s)$. However, not all meanings of a word might necessarily map to a single word sense due to the incompleteness of sense inventories. For example, a novel use of an existing word might be a combination of multiple existing senses of that word rather than a novel sense. Therefore, it is important to consider the sense distribution over word senses, $p(z_w|w, s)$, instead of only the most probable sense. Later in §5.1, we experimentally study the effect of using top-$k$ senses of a word in a given sentence.

### 3.2 Comparing Sense Distributions

Following the procedure described in §3.1, we independently compute the distributions $p(z_w|w, \mathcal{C}_1)$ and $p(z_w|w, \mathcal{C}_2)$ respectively from $\mathcal{C}_1$ and $\mathcal{C}_2$. Next, we compare those two distributions using different distance measures, $d(p(z_w|w, \mathcal{C}_1), p(z_w|w, \mathcal{C}_2))$, to determine whether the meaning of $w$ has changed between $\mathcal{C}_1$ and $\mathcal{C}_2$. For this purpose, we use five distance measures (i.e. Cosine, Chebyshev, Canberra, Bray-Curtis, Euclidean) and two divergence measures (i.e. Jensen-Shannon (JS), Kullback–Leibler (KL)) in our experiments. For computing distance measures, we consider each sense $z_w$ as a dimension in a vector space where the corresponding value is set to $p(z_w|w, \mathcal{C})$. The definitions of those measures are given in Appendix A.

## 4 Experiments

**Data and Evaluation Metrics:** We use the SemEval-2020 Task 1 dataset (Schlechtweg et al., 2020) to evaluate SCD of words over time for English, German, Swedish and Latin in two subtasks:

binary classification and ranking. In the classification subtask, the words in the evaluation set must be classified as to whether they have semantically changed over time. Classification **Accuracy** (i.e. percentage of the correctly predicted words in the set of test target words) is used as the evaluation metric for this task.

To predict whether a target word $w$ has its meaning changed, we use Bayesian optimisation to find a threshold on the distance, $d(p(z_w|w, \mathcal{C}_1), p(z_w|w, \mathcal{C}_2))$, between the sense distributions of $w$ computed from $\mathcal{C}_1$ and $\mathcal{C}_2$. Specifically, we use the Adaptive Experimentation Platform[4] to find the threshold that maximises the classification accuracy on a randomly selected held-out portion of words from the SemEval dataset, reserved for validation purposes. We found that using Bayesian optimisation is more efficient than conducting a linear search over the parameter space. We repeat this threshold estimation process five times and use the averaged parameter values in the remainder of the experiments.

In the ranking subtask, the words in the evaluation set must be sorted according to the degree of semantic change. Spearman's rank correlation coefficient ($\rho \in [-1, 1]$) between the human-rated gold scores and the induced ranking scores is used as the evaluation metric for this subtask. Higher $\rho$ values indicate better SCD methods.

Statistics of the data used in our experiments are shown in Table 4 in Appendix B. The English dataset includes two corpora from different centuries extracted from CCOHA (Alatrash et al., 2020). Let us denote the corpora collected from the early 1800s and late 1900s to early 2000s respectively by $C_1$ and $C_2$. For each language, its test set has 30-48 target words that are selected to indicate whether they have undergone a semantic change between the two time periods. These words are annotated by native speakers indicating whether their meanings have changed over time and if so the degree of the semantic change.

**Sense Embeddings:** We use two pre-trained sense embeddings in our experiments: LMMS (Loureiro and Jorge, 2019) and ARES (Scarlini et al., 2020b). For English monolingual experiments we use the 2048-dimensional

---

[4]https://ax.dev/

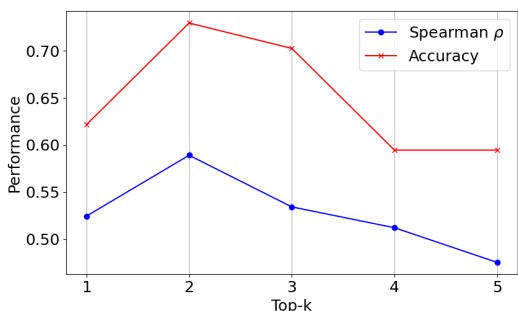

Figure 2: Spearman's $\rho$ and Accuracy on SemEval English held-out data when using the top-$k$ senses in the sense distribution with JS as the divergence.

LMMS[5] and ARES[6] embeddings computed using `bert-large-cased`,[7] which use WordNet sense ids. For the multilingual experiments, we use the 768-dimensional ARES embeddings computed using `multilingual-bert`,[8] which uses BabelNet sense ids. As a baseline method that does not use sense embeddings, we use the English WSD implementation in NLTK[9] to predict WordNet sense-ids for the target words, when computing the sense distributions, $p(z_w|s)$ in (1).

**Hardware and Hyperparameters:** We used a single NVIDIA RTX A6000 and 64 GB RAM in our experiments. It takes ca. 7 hours to compute semantic change scores for all target words in the 15.3M sentences in the SemEval datasets for all languages we consider. The only hyperparameter in SSCS is the threshold used in the binary classification task, which is tuned using Bayesian Optimisation. The obtained thresholds for the different distance measures are shown in Table 5 in Appendix B.

## 5 Results

### 5.1 Effect of Top-$k$ Senses

The correct sense of a word might not necessarily be ranked as the top-1 because of two reasons: (a) the sense embeddings might not perfectly encode all sense related information, and (b) the contextualised word embeddings that we use to compute the inner-product with sense embeddings might en-

---

[5] https://github.com/danlou/LMMS/tree/LMMS_ACL19
[6] http://sensembert.org/
[7] https://huggingface.co/bert-large-cased
[8] https://huggingface.co/bert-base-multilingual-cased
[9] https://www.nltk.org/howto/wsd.html

code information that is not relevant to the meaning of the target word in the given context. Therefore, there is some benefit of not strictly limiting the sense distribution only to the top-1 ranked sense, but to consider $k(\geq 1)$ senses for a target word in a given context. However, when we increase $k$, we will consider less likely senses of $w$ in a given context $s$, thereby introducing some noise in the computation of $p(z_w|w, \mathcal{C})$.

We study the effect of using more than one sense in a given context to compute the sense distribution of a target word. Specifically, we sort the senses in the descending order of $p(z_w|w, s)$, and select the top-$k$ ranked senses to represent $w$ in $s$. Setting $k = 1$ corresponds to considering the most probable sense, i.e. $\arg_{z_w \in \mathcal{Z}_w} \max p(z_w|w, s)$. In Figure 2, we plot the accuracy (for the binary classification subtask) and $\rho$ (for the ranking subtask) obtained using LMMS sense embeddings and JS divergence measure against $k$. For this experiment, we use a randomly held-out set of English words from the SemEval dataset. From Figure 2, we see that both accuracy and $\rho$ increase initially with $k$ up to a maximum at $k = 2$, and then start to drop. Following this result, we limit the sense distributions to the top 2 senses for the remainder of the experiments reported in this paper.

### 5.2 English Monolingual Results

Aida and Bollegala (2023) showed that the performance of an SCD method depends on the metric used for comparisons. Therefore, in Table 1 we show SCD results for English target words with different distance/divergence measures using LMMS, ARES sense embeddings against the NLTK WSD baseline. We see that LMMS sense embeddings coupled with JS divergence report the best performance for both the binary classification and ranking subtasks across all settings, whereas ARES sense embeddings with JS divergence obtain similar accuracy for the binary classification subtask. Comparing the WSD methods, we see that NLTK is not performing as well as the sense embedding-based methods (i.e. LMMS and ARES) in terms of Spearman's $\rho$ for the ranking subtask. Although NLTK matches the performance of ARES on the classification subtask, it is still below that of LMMS. Moreover, the best performance for NLTK for the classification subtask is achieved with multiple metrics such as Cosine, Bray-Curtis, Euclidean and KL. Therefore, we conclude that the sense embedding-

| | Metric | Spearman's $\rho$ | Accuracy |
|---|---|---|---|
| **NLTK** | Cosine | 0.007 | **0.595** |
| | Chebyshev | 0.301 | 0.541 |
| | Canberra | **0.423** | 0.568 |
| | Bray-Curtis | 0.175 | **0.595** |
| | Euclidean | 0.257 | **0.595** |
| | JS | 0.302 | 0.514 |
| | KL | 0.351 | **0.595** |
| **ARES** | Cosine | 0.149 | 0.595 |
| | Chebyshev | 0.080 | 0.568 |
| | Canberra | 0.447 | 0.649 |
| | Bray-Curtis | 0.485 | 0.622 |
| | Euclidean | 0.277 | 0.568 |
| | JS | **0.529** | **0.730**[†] |
| | KL | 0.233 | 0.622 |
| **LMMS** | Cosine | 0.274 | 0.622 |
| | Chebyshev | 0.124 | 0.568 |
| | Canberra | 0.502 | 0.405 |
| | Bray-Curtis | 0.541 | 0.676 |
| | Euclidean | 0.245 | 0.568 |
| | JS | **0.589**[†] | **0.730**[†] |
| | KL | 0.329 | 0.676 |

Table 1: Semantic Change Detection performance on SemEval 2020 Task 1 English dataset.

based sense distribution computation is superior to that of the NLTK baseline.

Among the different distance/divergence measures used, we see that JS divergence measure performs better than the other measures. In particular, for both ARES and LMMS, JS outperforms other measures for both subtasks, and emerges as the overall best metric for SCD. On the other hand, Cosine distance, which has been used as a baseline in much prior work on semantic change detection (Rosin et al., 2022) performs poorly for the ranking subtask although it does reasonably well on the classification subtask. Rosin et al. (2022) predicted semantic changes by thresholding the cosine distance. They used peak detection methods (Palshikar, 2009) to determine this threshold, whereas we use Bayesian optimisation methods.

### 5.3 English SCD Results

We compare SSCS against the following prior SCD methods on the SemEval-2020 Task 1 English data. Due to space limitations, further details of those methods are given in Appendix C.

**BERT + Time Tokens + Cosine** is the method proposed by Rosin et al. (2022) that fine-tunes pretrained BERT-base models using time tokens. **BERT + APD** was proposed by Kutuzov and Giulianelli (2020) that uses the averages pairwise cosine distance. Based on this insight, Aida and Bollegala (2023) evaluate the performance of **BERT + TimeTokens + Cosine** with the average pairwise cosine distance computed using pre-trained BERT-base as the MLM. **BERT+DSCD** is the sibling Distribution-based SCD (DSCD) method proposed by Aida and Bollegala (2023).

A **Temporal Attention** mechanism was proposed by Rosin and Radinsky (2022) where they add a trainable temporal attention matrix to the pretrained BERT models. Because their two proposed methods (fine-tuning with time tokens and temporal attention) are independent, Rosin and Radinsky (2022) proposed to use them simultaneously, which is denoted by **BERT + Time Tokens + Temporal Attention**. Yüksel et al. (2021) extended word2vec to create **Gaussian embeddings** (Vilnis and McCallum, 2015) for target words independently from each corpus

Rother et al. (2020) proposed Clustering on Manifolds of Contextualised Embeddings (**CMCE**) where they use mBERT embeddings with dimensionality reduction to represent target words, and then apply clustering algorithms to find the different sense clusters. CMCE is the current SoTA for the binary classification task. Asgari et al. (2020) proposed **EmbedLexChange**, which uses fasttext (Bojanowski et al., 2017) to create word embeddings from each corpora separately, and measures the cosine similarity between a word and a fixed set of pivotal words to represent a word in a corpus using the distribution over those pivots.

**UWB** (Pražák et al., 2020) learns separate word embeddings for a target word from each corpora and then use Canonical Correlation Analysis (CCA) to align the two vector spaces. UWB was ranked 1st for the binary classification subtask at the official SemEval 2020 Task 1 competition.

In Table 2, we compare our SSCS with JS using LMMS sense embeddings (which reported the best performance according to Table 1) against prior work. For prior SCD methods, we report performance from the original publications, without rerunning those methods. However, not all prior SCD methods evaluate on both binary classification and ranking subtasks as we do in this paper, which is indicated by N/A (not available) in Table 2. XL-

| Model | Accuracy | Spearman |
|---|---|---|
| BERT-base + TimeTokens + Cosine (Rosin et al., 2022) | N/A | 0.467 |
| BERT-base + APD (Kutuzov and Giulianelli, 2020) | N/A | 0.479 |
| BERT-base + Temporal Attention (Rosin and Radinsky, 2022) | N/A | 0.520 |
| BERT-base + TimeTokens + DSCD (Aida and Bollegala, 2023) | N/A | 0.529 |
| BERT-base + TimeTokens + Temporal Attention (Rosin and Radinsky, 2022) | N/A | 0.548 |
| Gaussian Embeddings (Yüksel et al., 2021) | 0.649 | 0.400 |
| CMCE (Rother et al., 2020) | **0.730** | 0.440 |
| EmbedLexChange (Asgari et al., 2020) | 0.703 | 0.300 |
| UWE (Pražák et al., 2020) | 0.622 | 0.365 |
| XL-LEXEME (Cassotti et al., 2023) (supervised) | N/A | **0.757** |
| SSCS (LMMS + JS) | **0.730** | 0.589 |

Table 2: Comparison against previously proposed SCD methods on the English data in SemEval-2020 Task 1.

LEXEME is a supervised SCD method that is the current SoTA on this dataset.

From Table 2, we see that SSCS obtains competitive results for both binary classification and ranking subtasks on the SemEval-2020 Task 1 English dataset, showing the effectiveness of word sense information for SCD. It matches the performance of CMCE for the binary classification subtask, while outperforming Temporal attention with Time Token fine-tuning (Rosin and Radinsky, 2022) on the ranking subtask.

Although models such as CMCE and EmbedLex-Change have good performance for the binary classification subtask, their performance on the ranking subtask is poor. Both of those methods learn static word embeddings for a target word *independently* from each corpus. Therefore, those methods must first learn comparable distributions before a distance measure can be used to calculate a semantic change score for a target word. As explained above, CMCE learns CCA-based vector space alignments, while EmbedLexChange uses the cosine similarity over a set of fixed pivotal words selected from the two corpora. Both vector space alignments and pivot selection are error prone, and add additional noise to SCD. On the other hand, SSCS uses the *same* MLM and sense embeddings on both corpora when computing the sense distributions, thus obviating the need for any costly vector space alignments.

Both TimeToken and Temporal Attention methods require retraining a transformer model (i.e. BERT models are used in the original papers). TimeToken prepends each sentence with a timestamp, thereby increasing the input length, which results in longer training times with transformer-based LLMs. On the other hand, Temporal Attention increases the number of parameters in the transformer as it uses an additional time-specific weight matrix. Interestingly, from Table 2 we see that SSCS outperforms both those methods convincingly despite not requiring any fine-tuning/retraining of the sense embeddings nor MLMs, which is computationally attractive.

SSCS (which is unsupervised) does not outperform XL-LEXEME (which is trained on WiC data) for the ranking subtask. In particular, we see a significant performance gap between XL-LEXEME and the rest of the unsupervised methods, indicating that future work on SCD should explore the possibility of incorporating some form a supervision to further improve performance. Although in SSCS, we used pre-trained static sense embeddings without any further fine-tuning, we could have used WiC data to select the classification threshold. During inference time, XL-LEXEME computes the average pair-wise cosine distance between the embeddings of sentences that contain the target word (which we are interested in predicting whether its meaning has changed over time), selected from each corpora. However, as already discussed in § 5.2, JS divergence outperforms cosine distance for SCD. Therefore, it would be an interesting future research direction would be to incorporate the findings from unsupervised SCD to further improve performance in supervised SCD methods.

### 5.4 Multilingual SCD Results

To evaluate the effectiveness of word sense distributions for detecting semantic change of words in other languages, we use the 768-dimensional ARES multilingual sense embed-

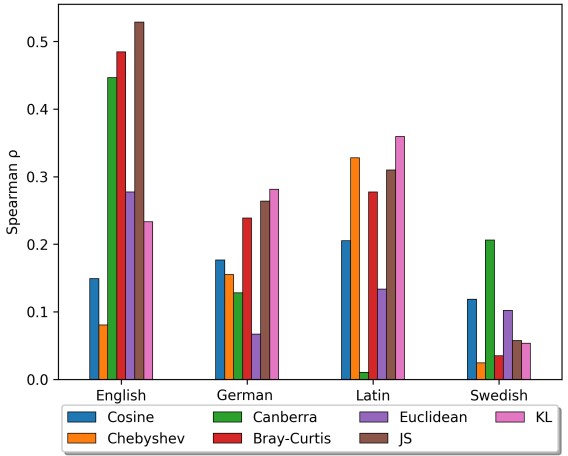

Figure 3: Multilingual semantic change detection of SSCS for ranking task in English, German, Latin and Swedish.

dings,[10] trained using BabelNet concept ids. We use `bert-base-multilingual-cased`[11] as the multilingual MLM in this evaluation because it is compatible with the ARES multilingual sense embeddings. For evaluations, we use the ranking subtask data in the SemEval 2020 Task 1 for German, Swedish and Latin.

In Figure 3 we compare $\rho$ values obtained using different distance/divergence measures. We see that JS performs best for English, KL for German and Latin, whereas Canberra for Swedish. Overall, the divergence-based measures (i.e. KL and JS) report better results than distance-based measures across languages, except in Swedish. Various factors affect the performance such as the coverage and sparseness of sense distributions, size of the corpora in each time period, and the number of test target words in each evaluation dataset. Therefore, it is difficult to attribute the performance differences across languages purely to the different distance/divergence measures used to compare the sense distributions.

The performance for non-English languages is much lower compared to that for English. This is due to three main reasons: (a) the limited sense coverage in BabelNet for non-English languages (especially Latin and Swedish in this case), (b) the accuracy of ARES sense embedding for German and Latin being lower compared to that for English, and (c) the multilingual contextualised embeddings

obtained from mBERT has poor coverage for Latin. Although more language-specialised MLMs are available such as GermanBERT[12], LatinBERT,[13] and SwedishBERT[14], we must have compatible sense embeddings to compute the sense distributions. Learning accurate multilingual sense embeddings is an active research area (Rezaee et al., 2021; Upadhyay et al., 2017) on its own and is beyond the scope of this paper which focuses on SCD.

## 5.5 Qualitative Analysis

Figure 4 shows example sense distributions and the corresponding JS divergence scores for the words *plane* (a word that has changed meaning according to SemEval annotators, giving a rating of 0.882) and *pin* (a word that has not changed its meaning, with a rating of 0.207) from the SemEval English binary classification subtask. We see that the two distributions for *plane* are significantly different from each other (the second peak at sense-id 5 vs. 6, respectively in $C_1$ and $C_2$), as indicated by a high JS divergence (i.e. 0.221). On the other hand, the sense distributions for *pin* are similar, resulting in a relatively smaller (i.e. 0.027) JS divergence. This result supports our claim that sense distributions provide useful clues for SCD of words.

In Table 3, we show the top- and bottom-8 ranked words according to their semantic change scores in the SemEval English dataset. We compare the ranks assigned to words according to SSCS against the NLTK baseline (used in Table 1) and DSCD (Aida and Bollegala, 2023). From Table 3 we see that for 6 (i.e. *plane, tip, graft, record, stab, head*) out of the top-8 ranked words with a semantic change between the corpora, SSCS assigns equal or lower ranks than either of NLTK or DSCD. Moreover, we see that SSCS assigns lower ranks to words that have not changed meaning across corpora.

As an error analysis, let us consider *risk*, which is assigned a higher rank (8) incorrectly by SSCS, despite not changing its meaning. Further investigations (see Appendix D) reveal that the sense distributions for *risk* computed from the two corpora are indeed very similar, except that $C_2$ has two additional senses not present in $C_1$. However, those additional senses are highly similar to ones

[10]http://sensembert.org/resources/ares_embedding.tar.gz
[11]https://huggingface.co/bert-base-multilingual-cased

[12]https://huggingface.co/bert-base-german-cased
[13]https://github.com/dbamman/latin-bert
[14]https://huggingface.co/KB/bert-base-swedish-cased

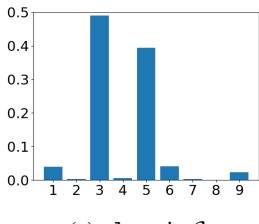
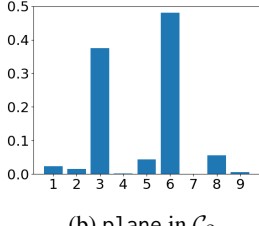
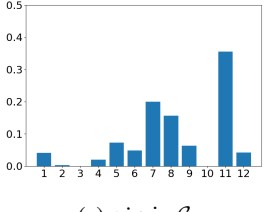
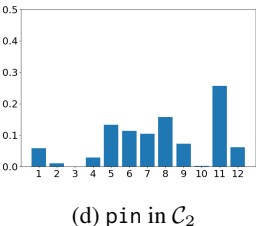

| (a) plane in $\mathcal{C}_1$ | (b) plane in $\mathcal{C}_2$ | (c) pin in $\mathcal{C}_1$ | (d) pin in $\mathcal{C}_2$ |

Figure 4: Sense distributions of `pin` and `plane` in the two corpora in SemEval 2020 Task 1 English dataset. In each subfigure, probability (in $y$-axis) is shown against the sense ids (in $x$-axis). The sense distributions of `plane` have changed across corpora, while that of `pin` remain similar. The human ratings for `plane` and `pin` are respectively 0.882 and 0.207, indicating that `plane` has changed its meaning between the two corpora, while `pin` has not. The JS divergence between the two sense distributions for `plane` is 0.221, while that for `pin` is 0.027.

| Word | Gold rank | $\Delta$ | NLTK rank | SSCS rank | DSCD rank |
|---|---|---|---|---|---|
| plane | 1 | ✓ | 2 | 1 | 15 |
| tip | 2 | ✓ | 17 | 6 | 7 |
| prop | 3 | ✓ | 24 | 17 | 4 |
| graft | 4 | ✓ | 23 | 4 | 36 |
| record | 5 | ✓ | 7 | 2 | 14 |
| stab | 6 | ✓ | 11 | 11 | 11 |
| bit | 7 | ✓ | 8 | 15 | 9 |
| head | 8 | ✓ | 14 | 10 | 28 |
| multitude | 30 | ✗ | 26 | 23 | 35 |
| savage | 31 | ✗ | 22 | 29 | 26 |
| contemplation | 32 | ✗ | 13 | 35 | 37 |
| tree | 33 | ✗ | 12 | 27 | 30 |
| relationship | 34 | ✗ | 37 | 31 | 34 |
| fiction | 35 | ✗ | 35 | 33 | 29 |
| chairman | 36 | ✗ | 27 | 34 | 33 |
| risk | 37 | ✗ | 31 | 8 | 21 |
| Spearman | 1.000 | | 0.462 | 0.589 | 0.529 |

Table 3: Ablation study on the top-8 semantically changed ($\Delta = ✓$) words with the highest degree of semantic change and the bottom-8 stable words ($\Delta = ✗$) with the lowest degree of semantic change. NLTK baseline performs WSD using NLTK's WSD functionality and uses KL to compare sense distributions. DSCD is proposed by Aida and Bollegala (2023) and approximates sibling embeddings using multivariate Gaussians. SSCS is our proposed method, which uses LMMS sense embeddings and JS as the distance metric.

present in $\mathcal{C}_1$ and imply the semantic invariance of *risk*. Explicitly incorporating sense similarity into SSCS could further improve its performance.

## 6 Conclusion

We proposed, SSCS, a sense distribution-based method for predicting the semantic change of a word from one corpus to another. SSCS obtains good performance among the unsuupervised methods for both binary classification and ranking subtasks for the English unsupervised SCD on the Se-

mEval 2020 Task 1 dataset. The experimental results highlight the effectiveness of using word sense distribution to detect semantic changes of words in different languages.

## Acknowledgements

Danushka Bollegala holds concurrent appointments as a Professor at University of Liverpool and as an Amazon Scholar. This paper describes work performed at the University of Liverpool and is not associated with Amazon.

## 7 Limitations

An important limitation in our proposed sense distribution-based SCD method is its reliance on sense labels. Sense labels could be obtained using WSD tools (as demonstrated by the use of NLTK baseline in our experiments) or by performing WSD using pre-trained static sense embeddings with contextualised word embeddings, obtained from pre-trained MLMs (as demonstrated by the use of LMMS and ARES in our experiments). Even if the WSD accuracy is not high for the top-1 predicted sense, SSCS can still accurately predict SCD because it uses the sense distribution for a target word, and not just the top-1 predicted sense. Moreover, SSCS uses the sense distribution of a target word over the entire corpus and not for a single sentence. Both WSD and sense embedding learning are active research topics in NLP (Bevilacqua and Navigli, 2020). We can expect the performance of WSD tools and sense embeddings to improve further in the future, which will further improve the SCD accuracy of SSCS.

Although we evaluated the performance of SSCS in German, Latin and Swedish in addition to English, this is still a limited set of languages. How-

ever, conducting a large-scale multilingual evaluation for SCD remains a formidable task due to the unavailability of human annotated semantic change scores/labels for words in many different languages. As demonstrated by the difficulties in data annotation tasks during the SemEval 2020 Task 1 on Unsupervised SCD (Schlechtweg et al., 2020), it is difficult to recruit native speakers for all languages of interest. Indeed for Latin it has been reported that each test word was annotated by only a single expert because it was not possible to recruit native speakers via crowd-sourcing for Latin, which is not a language in active usage. Therefore, we can expect similar challenges when evaluating SCD performance for rare or resource poor languages, with limited native speakers. Moreover, providing clear annotation guidelines for the semantic changes of a word in a corpus is difficult, especially when the semantic change happens gradually over a longer period of time.

The semantic changes of words considered in SemEval 2020 Task 1 dataset span relatively longer time periods, such as 50-200 years. Although it is possible to evaluate the performance of SCD methods for detecting semantic changes of words that happen over a longer period, it is unclear from the evaluations on this dataset whether SSCS can detect more short term semantic changes. For example, the word *corona* gained a novel meaning in 2019 with the wide spread of COVID-19 pandemic compared to its previous meanings (e.g. sun's corona rings and a beer brand). We believe that it is important for an SCD method to accurately detect such short term semantic changes, especially when used in applications such as information retrieval, where keywords associated with user interests vary over a relatively shorter period of time (e.g. seasonality related queries can vary over a few weeks to a few months).

## 8 Ethical Considerations

We considered the problem of SCD of words across corpora, sampled at different points in time. To evaluate our proposed method, SSCS, against previously proposed methods for SCD we use the publicly available SemEval 2020 Task 1 datasets. We are unaware of any social biases or other ethical issues reported regarding this dataset. Moreover, we did not collect, annotate, or distribute any datasets as part of this work. Therefore, we do not foresee any ethical concerns regarding our work.

Having said that, we would like to point out that we are using pre-trained MLMs and static sense embeddings in SSCS. MLMs are known to encode unfair social biases such as gender- or race-related biases (Basta et al., 2019). Moreover, Zhou et al. (2022) showed that static sense embeddings also encode unfair social biases. Therefore, it is unclear how such biases would affect the SCD performance of SSCS. On the other hand, some gender-related words such as *gay* have changed their meaning over the years (e.g. *offering fun and gaiety* vs. *someone who is sexually attracted to persons of the same sex*). The ability to correctly detect such changes will be important for NLP models to make fair and unbiased decisions and generate unbiased responses when interacting with human users in real-world applications.

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

## A  Distance and Divergence Measures

We describe the distance and divergence measures $d$ used in our experiments to compare the sense distributions $p_1(z) = p(z_w|w, \mathcal{C}_1)$ and $p_2(z) = p(z_w|w, \mathcal{C}_2)$ computed respectively from $\mathcal{C}_1$ and $\mathcal{C}_2$.

**Kullback-Liebler (KL) Divergence:**

$$\text{KL}(p_1||p_2) = \sum_{z \in \mathcal{Z}_w} (p_1(z)) \log\left(\frac{p_1(z)}{p_2(z)}\right) \quad (3)$$

**Jensen-Shannon (JS) Divergence:**

$$\text{JS}(p_1||p_2)$$
$$= \frac{1}{2}\text{KL}(p_1||q) + \frac{1}{2}\text{KL}(p_2||q) \quad (4)$$

Here,

$$q(z) = \frac{1}{2}(p_1(z) + p_2(z)) \quad (5)$$

**Bray-Curtis Distance:**

$$d(p_1, p_2) = \frac{\sum_{z \in \mathcal{Z}_w} |p_1(z) - p_2(z)|}{\sum_{z \in \mathcal{Z}_w} |p_1(z) + p_2(z)|} \quad (6)$$

**Canberra Distance:**

$$d(p_1, p_2) = \sum_{z \in \mathcal{Z}_w} \frac{|p_1(z) - p_2(z)|}{|p_1(z) + p_2(z)|} \quad (7)$$

**Chebyshev Distance:**

$$d(p_1, p_2) = \max_{z \in \mathcal{Z}_w} |p_1(z) - p_2(z)| \quad (8)$$

**Cosine Distance:**

$$d(p_1, p_2) = 1 - \frac{\sum_{z \in \mathcal{Z}_w} p_1(z)p_2(z)}{\sqrt{\sum_{z \in \mathcal{Z}_w} p_1(z)^2}\sqrt{\sum_{z \in \mathcal{Z}_w} p_1(z)^2}} \quad (9)$$

**Euclidean Distance:**

$$d(p_1, p_2) = \sqrt{\sum_z (p_1(z) - p_2(z))^2} \quad (10)$$

## B  Experimental Settings

Statistics of the SemEval 2020 Task 1 Unsupervised Semantic Change Detection Dataset is shown in Table 4. Thresholds found with Bayesian optimisation for the classification task are listed in Table 5. We used random seed of 42 while performing the Bayesian optimisation. From Table 5 we see that the classification thresholds that are learnt for different divergence/distance metrics are largely different.

## C  Previously Proposed SCD Methods

We compare SSCS against the following prior SCD methods on the SemEval-2020 Task 1 English data.

**BERT + Time Tokens + Cosine** : Rosin et al. (2022) performed fine-tuning of the published pretrained BERT-base models using time tokens. They add a time token (e.g. $<2023>$) to the beginning of a sentence. In the fine-tuning step, the models use two types of masked language modelling objectives: 1) predicting the masked time tokens from given contexts, and 2) predicting the masked tokens from given contexts with time tokens. They make predictions with the average distance of the target token probabilities or the cosine distance of the average sibling embeddings. According to their results, the cosine distance achieves better performance than the average distance of the probabilities. Therefore, we use cosine distance as the distance metric for this method.

| Language | Time Period | #Targets | #Sentences | #Tokens | #Types |
|---|---|---|---|---|---|
| English | 1810–1860 | 37 | 254k | 6.5M | 87k |
|  | 1960–2010 |  | 354k | 6.7M | 150k |
| German | 1800–1899 | 48 | 2.6M | 70.2M | 1.0M |
|  | 1946–1990 |  | 3.5M | 72.3M | 2.3M |
| Latin | B.C. 200–0 | 40 | 96k | 1.7M | 65k |
|  | 0-2000 |  | 463k | 9.4M | 253k |

Table 4: Statistics of the SemEval 2020 Task 1 Unsupervised Semantic Change Detection Dataset.

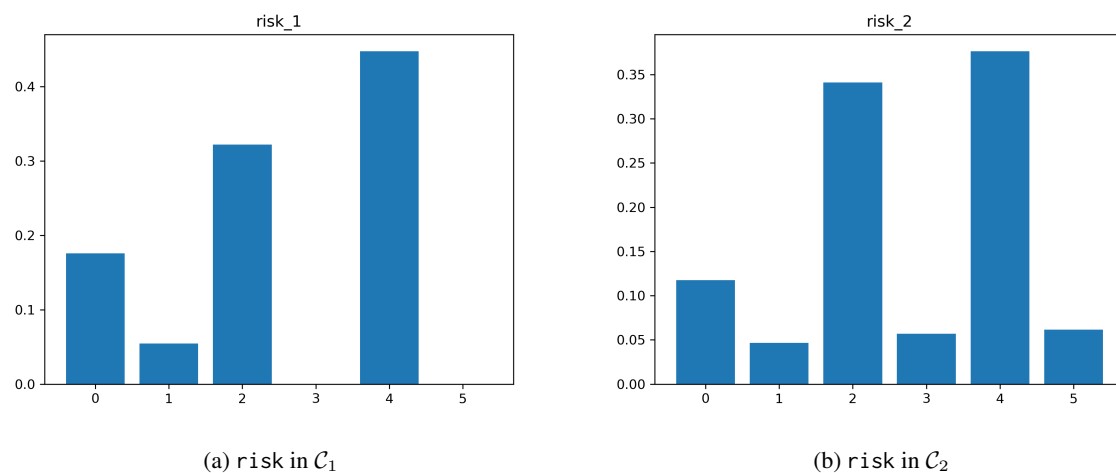

(a) `risk` in $\mathcal{C}_1$

(b) `risk` in $\mathcal{C}_2$

Figure 5: Sense distributions computed for the word `risk` from $\mathcal{C}_1$ and $\mathcal{C}_2$.

**BERT + APD** : Kutuzov and Giulianelli (2020) report that the average pairwise cosine distance outperforms the cosine distance. Based on this insight, Aida and Bollegala (2023) evaluate the performance of **BERT + TimeTokens + Cosine** with the average pairwise cosine distance computed using pre-trained `BERT-base` as the MLM.

**BERT+DSCD** : Aida and Bollegala (2023) proposed a Distribution-based Semantic Change Detection (DSCD), considering distributions of sibling embeddings (*sibling distribution*). During prediction, they sample an equal number of target word vectors from the sibling distribution (approximated by a multivariate diagonal Gaussian) for each time period and calculate the average distance. They report that the Chebyshev distance function achieves the best performance.

**Temporal Attention** : Rosin and Radinsky (2022) proposed a temporal attention mechanism, where they add a trainable temporal attention matrix to the pretrained BERT models. Subsequently, additional training is performed on the target cor-

pus. They use the cosine distance following their earlier work (Rosin et al., 2022).

**BERT + Time Tokens + Temporal Attention** : Because their two proposed methods (fine-tuning with time tokens and temporal attention) are independent, Rosin and Radinsky (2022) proposed to use them simultaneously. They also use the cosine distance for prediction and report that this model achieves current state-of-the-art performance in semantic change detection.

**Gaussian Embeddings:** Yüksel et al. (2021) extend word2vec to create Gaussian embeddings (Vilnis and McCallum, 2015) for target words independently in each corpus. They then learn a mapping between the two vector spaces for computing a semantic change score.

**CMCE:** Rother et al. (2020) proposed Clustering on Manifolds of Contextualised Embeddings (CMCE) where they use mBERT embeddings with dimensionality reduction to represent target words, and then apply clustering algorithms to find the different sense clusters. CMCE is the current SoTA

| | Metric | Threshold |
|---|---|---|
| **NLTK** | Cosine | 0.132 |
| | Chebyshev | 146.808 |
| | Canberra | 0.829 |
| | Bray-Curtis | 0.046 |
| | Euclidean | 63.024 |
| | JS | 0.0285 |
| | KL | 0.024 |
| **ARES** | Cosine | 0.3238 |
| | Chebyshev | 212.995 |
| | Canberra | 0.901 |
| | Bray-Curtis | 0.036 |
| | Euclidean | 509.011 |
| | JS | 0.080 |
| | KL | 0.108 |
| **LMMS** | Cosine | 0.052 |
| | Chebyshev | 206.235 |
| | Canberra | 0 |
| | Bray-Curtis | 0.103 |
| | Euclidean | 331.680 |
| | JS | 0.117 |
| | KL | 0.065 |

Table 5: Thresholds found with Bayesian optimisation for the classification task on English datasets.

for the binary classification task.

**EmbedLexChange:** Asgari et al. (2020) used fasttext (Bojanowski et al., 2017) to create word embeddings from each corpora separately, and measure the cosine similarity between a word and a selected fixed set of pivotal words to represent a word in a corpus using the distribution over those pivots. They used KL divergence to measure the similarity between the two distributions associated with a target word to compute a semantic change score.

**UWB:** Pražák et al. (2020) learnt separate word embeddings for a target word from each corpora and then use Canonical Correlation Analysis (CCA) to align the two vector spaces. They use the average cosine similarity between the two embeddings over all target words as the threshold for predicting semantic changes of words. UWB was ranked 1st for the binary classification subtask at the official SemEval 2020 Task 1 competition.

## D  Error Analysis

As a concrete example of computing semantic change scores using sense distributions where SSCS assigned a low JS divergence value using LMMS incorrectly to a word that had not changed its meaning, we consider the word *risk*. The sense distributions, $p(z_w|w, C_1)$ and $p(z_w|w, C_2)$, of *risk* in the two corpora, respectively $\mathcal{C}_1$ and $\mathcal{C}_2$ are shown in Figure 5. The word *risk* is considered as a noun in the SemEval 2020 Task 1 dataset, and the annotator-assigned semantic change score is 0, indicating that it has not changed meaning between the two corpora. Looking at the sense distributions in Figure 5, we see that they are almost similar with two peaks at sense-ids 2 and 4. However, two additional senses, corresponding sense-id 3 ('risk%1:07:02::', *risk of exposure, the probability of being exposed to an infectious agent*) and sense-id 5 ('risk%1:07:01::', *expose to a chance of loss or damage*) can be observed in Figure 5b in $\mathcal{C}_2$. According to the sense definitions in the WordNet for those two senses, they are very similar (i.e. expressing different types of risks), which could be considered to be a case where the meaning of *risk* remains largely the same in the two corpora. Because of this reason the JS divergence score between the two distributions tends to be higher at 0.2139, giving it a lower rank (rank 8 among 37 words in Table 3, indicating that *risk* has changed its meaning). This is a typical example where although a word might take different senses in different corpora (or within different sentences in the same corpora), some of those senses could be highly similar and could be mapped to the same meaning for the purpose of SCD.