# OpenReview forum: "A Word Sense Distribution-based approach for Semantic Change Prediction"
_EMNLP/2023/Conference — EMNLP 2023 Findings_

### Official Review · Reviewer_dbcW · 2023-08-05

**Soundness:** 3

**Excitement:**

3: Ambivalent: It has merits (e.g., it reports state-of-the-art results, the idea is nice), but there are key weaknesses (e.g., it describes incremental work), and it can significantly benefit from another round of revision. However, I won't object to accepting it if my co-reviewers champion it.

**Missing References:**

Related Works:
Rachinskiy, M., & Arefyev, N. (2021). Zeroshot crosslingual transfer of a gloss language model for semantic change detection. In Computational linguistics and intellectual technologies: Papers from the annual conference Dialogue (Vol. 20, pp. 578-586).

Maxim Rachinskiy and Nikolay Arefyev. 2021. GlossReader at SemEval-2021 Task 2: Reading Definitions Improves Contextualized Word Embeddings. In Proceedings of the 15th International Workshop on Semantic Evaluation (SemEval-2021), pages 756–762, Online. Association for Computational Linguistics.

SOTA Results:
Pierluigi Cassotti, Lucia Siciliani, Marco DeGemmis, Giovanni Semeraro, and Pierpaolo Basile. 2023. XL-LEXEME: WiC Pretrained Model for Cross-Lingual LEXical sEMantic changE. In Proceedings of the 61st Annual Meeting of the Association for Computational Linguistics (Volume 2: Short Papers), pages 1577–1585, Toronto, Canada. Association for Computational Linguistics.

**Paper Topic And Main Contributions:**

This paper introduces the Sense-based Semantic Change Score (SSCS), an unsupervised method for detecting semantic changes in words over time. By using pre-trained static sense embeddings to annotate occurrences of target words with sense labels and computing their distributions in different corpora, SSCS quantifies semantic change through divergence or distance measures. The sense embeddings are obtained using LMMS and ARES pre-trained embeddings for the English monolingual and ARES multilingual pre-trained embeddings for German and Latin.

**Questions For The Authors:**

A) Why was Swedish (SemEval 2020 Task 1) omitted from the experiments?
B) Why did you not use the RuShiftEval dataset in the evaluation, allowing you to evaluate your model against the similar "GlossReader" model?
C) Why have you not tried state-of-the-art WSD models, such as ESC? (ESC: Redesigning WSD with extractive sense comprehension)

**Reasons To Accept:**

The paper presents a significant contribution by introducing the application of Word Sense Disambiguation (WSD) tools in the diachronic domain for English, Latin, and German languages. This novel approach opens up exciting avenues for studying sense-aware semantic changes over time, largely unexplored in the literature. The qualitative analysis provided in the paper offers fresh insights and reveals different perspectives from existing research, highlighting the uniqueness of the proposed method. While the paper acknowledges the potential of explainability offered by WSD tools, further exploring this aspect could lead to a more in-depth understanding of the results.

**Reasons To Reject:**

The use of different measures, such as cosine distance and Jensen-Shannon Divergence (JSD), is not adequately justified. Without a clear explanation of the advantages brought by each measure, the experiments appear arbitrary, resembling a brute-force approach to improve results rather than a well-reasoned methodology. Additionally, the paper lacks essential references to the model with state-of-the-art results, and other related works still need to be included (See Missing References Section). Moreover, the rationale behind using Bayesian Optimization needs to be better-supported, especially considering that the threshold search problem is relatively simple and involves only one dimension. The paper should have emphasized and highlighted the results obtained maximizing the accuracy, ensuring a fair and comprehensive comparison with other approaches.

**Reproducibility:**

4: Could mostly reproduce the results, but there may be some variation because of sample variance or minor variations in their interpretation of the protocol or method.

**Reviewer Confidence:**

5: Positive that my evaluation is correct. I read the paper very carefully and I am very familiar with related work.

**Typos Grammar Style And Presentation Improvements:**

Row 122: Typo in the Tahmasebi reference Tahmasebia et al. 2021 -> Tahmasebi et al. 2021

---

> ### Author Rebuttal · Authors · 2023-08-27
>
> **RR1:** *The use of different measures, such as cosine distance and Jensen-Shannon Divergence (JSD), is not adequately justified. Without a clear explanation of the advantages brought by each measure, the experiments appear arbitrary, resembling a brute-force approach to improve results rather than a well-reasoned methodology. *
>
> **Response:** Different divergence and distance measures have been used already in much prior work on semantic change detection. As shown by Aida and Bollegala (ACL 2023), different distance/divergence measures perform differently under different settings (e.g. MLMs, languages, frequency of the words etc.). Therefore, we consider it would be both important and informative for the readers to learn about the performance of our proposed method when used with different divergence/distance measures. This is our motivation to experiment with different divergence/distance measures in Section 5.4 (Multilingual analysis). We will include this motivation in the final version of this paper.
>
> **RR2:** Moreover, the rationale behind using Bayesian Optimization needs to be better-supported, especially considering that the threshold search problem is relatively simple and involves only one dimension.
>
> **Response:** Bayesian optimisation is a simple yet highly efficient technique for estimating thresholds for classification tasks as popularly used in NLP [https://aclanthology.org/2023.findings-acl.550/] For example, unlike linear search, which requires us to pre-define the range and the step for increment during the search, Bayesian optimisation takes care of all these settings.
>
> **Q1:** *Why was Swedish (SemEval 2020 Task 1) omitted from the experiments?*
>
> **Response:**  We could not find publicly available pretrained sense embeddings for Swedish, which is the reason why we could not evaluate for Swedish.
>
> **Q2:** *Why did you not use the RuShiftEval dataset in the evaluation, allowing you to evaluate your model against the similar "GlossReader" model?*
>
> **Response:** We were not aware of this dataset at the time of submission because it has not been used by any prior work on SemEval 2020 Task 1 lexical semantic change detection.  Moreover, this dataset is limited to the Russian language. However, we can easily evaluate our proposed method for Russian semantic change detection using this dataset in the final version, which should further strengthen the multilingual results in this paper.
>
> **Q3:**  *Why have you not tried state-of-the-art WSD models, such as ESC? (ESC: Redesigning WSD with extractive sense comprehension)*
>
> **Response:** Please note that our proposed method does not require WSD tools. It uses only pretrained sense embeddings. WSD is used only as a baseline in the evaluations. One can use any WSD tool, including ESC, for this purpose.

---

### Official Review · Reviewer_jdV6 · 2023-08-09

**Soundness:** 1

**Excitement:**

1: Poor: I cannot identify the contributions of this paper, or I believe the claims are not sufficiently backed up by evidence. I would fight to have it rejected.

**Missing References:**

- Andrey Kutuzov, Erik Velldal, and Lilja Øvrelid. 2022. Contextualized embeddings for semantic change detection: Lessons learned. In Northern European Journal of Language Technology, Volume 8, Copenhagen, Denmark. Northern European Association of Language Technology.
- Severin Laicher, Sinan Kurtyigit, Dominik Schlechtweg, Jonas Kuhn, and Sabine Schulte im Walde. 2021. Explaining and Improving BERT Performance on Lexical Semantic Change Detection. In Proceedings of the 16th Conference of the European Chapter of the Association for Computational Linguistics: Student Research Workshop, pages 192–202, Online. Association for Computational Linguistics.
- Francesco Periti, Alfio Ferrara, Stefano Montanelli, and Martin Ruskov. 2022. What is Done is Done: an Incremental Approach to Semantic Shift Detection. In Proceedings of the 3rd Workshop on Computational Approaches to Historical Language Change, pages 33–43, Dublin, Ireland. Association for Computational Linguistics.
- Montanelli, Stefano, and Francesco Periti. "A Survey on Contextualised Semantic Shift Detection." arXiv preprint arXiv:2304.01666 (2023).
- Matej Martinc, Syrielle Montariol, Elaine Zosa, and Lidia Pivovarova. 2020. Capturing Evolution in Word Usage: Just Add More Clusters? In Companion Proceedings of the Web Conference 2020 (WWW '20). Association for Computing Machinery, New York, NY, USA, 343–349. https://doi.org/10.1145/3366424.3382186
- Maxim Rachinskiy and Nikolay Arefyev. 2022. GlossReader at LSCDiscovery: Train to Select a Proper Gloss in English – Discover Lexical Semantic Change in Spanish. In Proceedings of the 3rd Workshop on Computational Approaches to Historical Language Change, pages 198–203, Dublin, Ireland. Association for Computational Linguistics.
- Rachinskiy, M., & Arefyev, N. (2021). Zeroshot crosslingual transfer of a gloss language model for semantic change detection. In Computational linguistics and intellectual technologies: Papers from the annual conference Dialogue (Vol. 20, pp. 578-586).

**Paper Topic And Main Contributions:**

The paper proposes a novel approach to Lexical Semantic Change Detection called Sense-based Semantic Change Score (SSCS). This approach leverages sense embedding representations derived from pretrained models to differentiate between the multiple meanings a word conveys in two different corpora. Then, it quantifies the semantic change of the word by computing the divergence/distance in the time-specific sense distributions. The reported results indicate that SSCS outperform other state-of-the-art work in detecting semantic change over the SemEval 2020 Task 1 benchmark.

**Reasons To Accept:**

The paper is noteworthy for being among the first works to propose the use of Sense Embeddings derived from pretrained contextualized model for detecting lexical semantic changes. The paper is well-written and effectively presents its topic in a clear and understandable manner. The methodology and its underlying setting are clearly presented, the technical quality is appropriate.

**Reasons To Reject:**

The paper's novelty appears to be somewhat limited. One compelling reason to consider this paper is the authors' claim of outperforming existing methods (specifically for English). However, the paper's comparison isn't exhaustive, as there are other state-of-the-art approaches that demonstrate superior performance. For instance, when examining the ranking task for the SemEval English dataset:
- [1] presents results indicating that the smaller the model, the better the performance. In particular, [1] reports scores of .520 with BERT-base, .589 with BERT-small, and .627 with BERT-tiny (thus, the highest result is .627, rather than .520 - as indicated in the proposed paper).
- [2] and [3] report a score of .605 using ELMo.
- [11] reports .757 (even if it's supervised)
- Additionally, other works, although not referenced in this paper, exhibit comparable results such as .571 [4] and .512 [5]. While SSCS achieved a sligltly superior score (i.e. 589), it could be argued that for a relatively small set of target words like in the SemEval English dataset, such marginal differences hold negligible significance.
For a comprehensive comparison over different languages, I recommend referring to [6].

The major novelty of the paper is the use of sense embeddings in semantic change detection. Instead, it's worth noting that various unsupervised methods already exist, focusing on distinguishing word meanings (e.g. by Unsupervised clustering such as [5] and [7]; or by the use of Supervision through word sense induction [8], or Gloss-based WSD systems [9]) that calculate semantic changes by measuring divergence in sense distributions (typically JSD).

Introducing SSCS would be more valuable if it leads to improved results. While it's true, as mentioned in the paper, that unsupervised methods based on embedding clustering are not tied to specific sense inventories, it's important to recognize that such sense inventories might not be widely available, particularly for languages with limited resources. This is especially relevant in scenarios involving Historical texts, like those found in the SemEval datasets.

Exploring a qualitative comparison between the sense identification made by SSCS and other (un)supervised clustering methods could be extremely valuable. This comparison could serve to clearly highlight the superiority of the SSCS approach and sense embeddings in the realm of Lexical Semantic Change Detection by using word embeddings.

Other minor points:
- The abstract states that the method is computationally efficient, but there is no reference in the paper to clarify this statement and demonstrate this feature compared to other approaches.
- The abstract states that the method significantly outperforms state-of-the-art (SOTA) approaches, but it only performs better in English, specifically for the mentioned work, and the improvement is of a small magnitude, i.e. 0.041.
- A comparison against previously proposed SCD methods (as presented in Table 2 for English) should also be provided for the other languages considered.
- The author should provide a more thorough explanation of the intuition behind conducting experiments with various distance/divergence measures.
- The authors claim that the performance in German and Latin is much lower compared to that in English, primarily due to the limited sense coverage in BabelNet for non-English languages. Given that the zero-shot cross-lingual transferability of contextualized embedding models has recently proven to be effective in Semantic Change Detection [9,10], it would be beneficial for the authors to further investigate this statement. For instance, they could explore finetuning the model or comparing their approach to other works, taking into consideration benchmarks for languages like Spanish and Russian

References:
- [1] Guy D. Rosin and Kira Radinsky. 2022. Temporal Attention for Language Models. In Findings of the Association for Computational Linguistics: NAACL 2022, pages 1498–1508, Seattle, United States. Association for Computational Linguistics.
- [2] Andrey Kutuzov, Erik Velldal, and Lilja Øvrelid. 2022. Contextualized embeddings for semantic change detection: Lessons learned. In Northern European Journal of Language Technology, Volume 8, Copenhagen, Denmark. Northern European Association of Language Technology.
- [3] Andrey Kutuzov and Mario Giulianelli. 2020. UiO-UvA at SemEval-2020 Task 1: Contextualised Embeddings for Lexical Semantic Change Detection. In Proceedings of the Fourteenth Workshop on Semantic Evaluation, pages 126–134, Barcelona (online). International Committee for Computational Linguistics.
- [4] Severin Laicher, Sinan Kurtyigit, Dominik Schlechtweg, Jonas Kuhn, and Sabine Schulte im Walde. 2021. Explaining and Improving BERT Performance on Lexical Semantic Change Detection. In Proceedings of the 16th Conference of the European Chapter of the Association for Computational Linguistics: Student Research Workshop, pages 192–202, Online. Association for Computational Linguistics.
- [5] Francesco Periti, Alfio Ferrara, Stefano Montanelli, and Martin Ruskov. 2022. What is Done is Done: an Incremental Approach to Semantic Shift Detection. In Proceedings of the 3rd Workshop on Computational Approaches to Historical Language Change, pages 33–43, Dublin, Ireland. Association for Computational Linguistics.
- [6] Montanelli, Stefano, and Francesco Periti. "A Survey on Contextualised Semantic Shift Detection." arXiv preprint arXiv:2304.01666 (2023).
- [7] Matej Martinc, Syrielle Montariol, Elaine Zosa, and Lidia Pivovarova. 2020. Capturing Evolution in Word Usage: Just Add More Clusters? In Companion Proceedings of the Web Conference 2020 (WWW '20). Association for Computing Machinery, New York, NY, USA, 343–349. https://doi.org/10.1145/3366424.3382186
- [8] Renfen Hu, Shen Li, and Shichen Liang. 2019. Diachronic Sense Modeling with Deep Contextualized Word Embeddings: An Ecological View. In Proceedings of the 57th Annual Meeting of the Association for Computational Linguistics, pages 3899–3908, Florence, Italy. Association for Computational Linguistics.
- [9] Maxim Rachinskiy and Nikolay Arefyev. 2022. GlossReader at LSCDiscovery: Train to Select a Proper Gloss in English – Discover Lexical Semantic Change in Spanish. In Proceedings of the 3rd Workshop on Computational Approaches to Historical Language Change, pages 198–203, Dublin, Ireland. Association for Computational Linguistics.
- [10] Rachinskiy, M., & Arefyev, N. (2021). Zeroshot crosslingual transfer of a gloss language model for semantic change detection. In Computational linguistics and intellectual technologies: Papers from the annual conference Dialogue (Vol. 20, pp. 578-586).
- [11] Pierluigi Cassotti, Lucia Siciliani, Marco DeGemmis, Giovanni Semeraro, and Pierpaolo Basile. 2023. XL-LEXEME: WiC Pretrained Model for Cross-Lingual LEXical sEMantic changE. In Proceedings of the 61st Annual Meeting of the Association for Computational Linguistics (Volume 2: Short Papers), pages 1577–1585, Toronto, Canada. Association for Computational Linguistics.

**Reproducibility:**

4: Could mostly reproduce the results, but there may be some variation because of sample variance or minor variations in their interpretation of the protocol or method.

**Reviewer Confidence:**

5: Positive that my evaluation is correct. I read the paper very carefully and I am very familiar with related work.

**Typos Grammar Style And Presentation Improvements:**

- row 077: temporal shift -> I would like to suggest adopting a more widely established terminology, as seen in prior research, such as "lexical semantic change" or "semantic shift".
- row 135: two groups: (a) methods that compare word/context clusters, and (b) methods that compare embeddings of the target words computed from different corpora sampled at different time periods -> I would like to suggest adopting a more widely established terminology, as seen in prior research, such as sense-based and form-based approaches
- row 500: esbalishing -> establishing
- row 512: measures -> measure
- row 518: adds -> add

---

> ### Author Rebuttal · Authors · 2023-08-27
>
> **RR1:** *The paper's novelty appears to be somewhat limited. *
>
> **Response:** We respectfully disagree. As you have already stated in the reasons for acceptance our work is a pioneering effort in incorporating sense embeddings for semantic change detection. Moreover, the proposed method already obtains SoTA for English on SemEval 2020 Task 1 benchmark, although not for other languages. Nevertheless, according to ACL reviewer guidelines [https://2023.aclweb.org/blog/review-acl23/], papers should not be rejected purely based on not being able to obtain SoTA.
>
> **RR2:** *Unsupervised clustering methods for distinguishing word meanings already exists, thus this paper lacks novelty*
>
> **Response:** Please note that “distinguishing word meanings” and “measuring semantic changes of words” are related but vastly different tasks. In the latter (which this paper is about), word sense information has never been used before as a signal, where the novelty of our work lies upon. On the other hand, unsupervised sense induction methods can be further helpful to improve the performance of our proposed method especially when human-created sense inventories are not available for a language of interest.
>
> **RR3:** *Exploring a qualitative comparison between the sense identification made by SSCS and other (un)supervised clustering methods could be extremely valuable.*
>
> **Response:** We agree it would be interesting to see whether we can use unsupervised clustering methods for sense induction (particularly useful when human-compiled sense inventories do not exist for learning sense embeddings for resource poor languages) with the proposed method. However, this is beyond the scope of the current paper and better done in an extended journal version of this 8 page conference paper because our main claim/contribution in this paper is “given a sense embedding, how to compute a semantic change detection score”.
>
> **RR3:** *The abstract states that the method is computationally efficient, but there is no reference in the paper to clarify this statement and demonstrate this feature compared to other approaches.*
>
> **Response:** As stated in Section 3.5 (Comparisons against SoTA), our proposed method does not require training transformer models with specialised temporal attention mechanisms [Rosin and Radinsky 2022] nor require fine-tuning existing MLMs on corpora sampled from the time points [Kuzutov and Guilianelli 2020]. For these reasons, we consider our proposed method to be computationally efficient compared to prior work on semantic change detection.
>
> **RR4:** *The abstract states that the method significantly outperforms state-of-the-art (SOTA) approaches, but it only performs better in English, specifically for the mentioned work, and the improvement is of a small magnitude, i.e. 0.041.*
>
> **Response:** Although the improvement might appear to be small,  the proposed method obtains SoTA for English on SemEval 2020 Task 1 dataset. This should not be used as a reason for rejection according to the ACL reviewer guidelines.
>
> **RR5:** *A comparison against previously proposed SCD methods (as presented in Table 2 for English) should also be provided for the other languages considered.*
>
> **Response:**  This information is already available in prior work, and we can easily include such a Table with the extra page allowed for the camera ready version if accepted.
>
> **RR6:** *The author should provide a more thorough explanation of the intuition behind conducting experiments with various distance/divergence measures.*
>
> **Response:** Different divergence and distance measures have been used already in much prior work on semantic change detection. As shown by Aida and Bollegala (ACL 2023), different distance/divergence measures perform differently under different settings (e.g. MLMs, languages, frequency of the words etc.). Therefore, we consider it would be both important and informative for the readers to learn about the performance of our proposed method when used with different divergence/distance measures. This is our motivation to experiment with different divergence/distance measures in Section 5.4 (Multilingual analysis). We will include this motivation in the final version of this paper.
>
> **Suggestion:** *Evaluations on Spanish and Russian…*
>
> **Response:** Thank you for the suggestions on the evaluation for Spanish and Russian. Although this would be beyond the scope of the conference version of this paper, we plan to conduct evaluations on these languages in an extended journal version to further evaluate the multilingual applicability of the proposed method.

---

### Official Review · Reviewer_LWUC · 2023-08-09

**Typos Grammar Style And Presentation Improvements:** The author names in the reference to …
**Soundness:** 4

**Excitement:**

4: Strong: This paper deepens the understanding of some phenomenon or lowers the barriers to an existing research direction.

**Paper Topic And Main Contributions:**

This paper tackles the task of semantic change detection (by way of the standard SemEval 2020 Task 1). It introduces a new method that involves word sense annotation, and then comparing the distributions of senses of target word types between corpora from different time periods. The word sense annotation task is done by using reference sense embeddings and comparing contextual embeddings from the corpus to these sense embeddings. Various different distance and divergence measures are tried to compare the distributions, and different sense embeddings are tried. The method beat the state-of-the-art result for this task, and the authors compare the metrics to various other recent approaches to this task. The task is done for English, German and Latin.

**Reasons To Accept:**

The paper contributes a methodological innovation in the area of semantic change detection and beats the state of the art at the time of this paper's submission. It is demonstrated for several languages, and it is methodologically thorough, trying different sense embeddings and taking the top-k senses from the classification rather than just one.

The method is relatively transparent due to its reliance on word senses - an interested scholar can easily inspect which parts (senses) of the distribution have changed, as is also demonstrated in the paper.

**Reasons To Reject:**

The reliance on sense labels means that the method might be less useful in specific domains, or with new words and senses such as the case of novel slang terms on social media.

Results are only shown for a rather large time gap (different centuries). This may limit the practical significance of the work, as the authors note, and it is not clear that the question of temporal generalization raised in the introduction can be addressed.

A rather elaborate chain of resources is necessary to carry out this task, limiting its applicability to under-resourced languages: there should be a reliable and broad sense inventory, pretrained sense embeddings and contextualized embeddings (though apparently just using mBERT performs well).

**Reproducibility:**

4: Could mostly reproduce the results, but there may be some variation because of sample variance or minor variations in their interpretation of the protocol or method.

**Reviewer Confidence:**

3: Pretty sure, but there's a chance I missed something. Although I have a good feel for this area in general, I did not carefully check the paper's details, e.g., the math, experimental design, or novelty.

---

> ### Author Rebuttal · Authors · 2023-08-27
>
> **RR1:** *The reliance on sense labels means that the method might be less useful in specific domains, or with new words and senses such as the case of novel slang terms on social media.*
>
> **Response:** Please note that we have discussed this point already in the Limitations section in the paper. Research on WSD is active and on-going and more accurate WSD tools are being developed for low-resource languages and specialised domains [Gogoi and Baruah 2022, Zhang+2022].
>
> On the other hand, please note that our proposed method does not depend on actual sense labels but can operate using "sense clusters". Specifically, provided with a method that can "discriminate" different senses of a word ("disambiguation" is not required), we can represent a word by a distribution of sense cluster ids. Moreover, sense clusters could be replaced by “topics” that are automatically generated via a topic modelling method such as LDA [Blei+ 2003], in a “sense induction” manner, thus obviating the need for sense labels at all. The feasibility of these extensions are deferred to future work, which would be inspired by the current paper.
>
> **RR2:** *Results are only shown for a rather large time gap*
>
> **Response:**  Please note that this is not a limitation of our proposed method but rather a limitation of benchmark datasets that would enable us to test the proposed method (as well as all prior work on lexical semantic change detection) for shorter time durations. The SemEval 2020 Task 1 dataset that we use in our experiments (and considered as the standard evaluation benchmark for lexical semantic change detection in much prior work) lists words that change their meaning over a relatively longer period of time.
>
> **RR3:** *An elaborate chain of resources is necessary to carry out this task*
>
> **Response:** Although the proposed method does depend on WSD tools and contextualised word embeddings obtained from MLMs, we disagree that these are hard-to-find resources in NLP nowadays. For example, WSD tools have been developed for many languages [Elayeb 2019, Orlando+2021, Zhang+ 2022, Lendvai 2022, Su+ 2022] and pretrained monolingual and multilingual MLMs are available [Liu+ 2019, Devlin+ 2019, Lan+ 2020, Conneau+ 2020, Barbieri+ 2022]. Moreover, the NLP community is actively working on WSD and MLMs, which means that the availability and quality of these resources will further improve in the future, thus having a positive domino effect on the performance of our proposed method.

---

### Meta-Review · Area_Chair_icfk · 2023-09-19

**Recommendation:** 2

**Metareview:**

The content and results are interesting but there appear to be many missing references.

---

### Decision · Program_Chairs · 2023-10-07

**Decision:**

Accept-Findings

**Comment:**

The content and results are interesting but there appear to be many missing references.